# Effects of Omega-3 Supplementation Alone and Combined with Resistance Exercise on Skeletal Muscle in Older Adults: A Systematic Review and Meta-Analysis

**DOI:** 10.3390/nu14112221

**Published:** 2022-05-26

**Authors:** Stephen M. Cornish, Dean M. Cordingley, Keely A. Shaw, Scott C. Forbes, Taylor Leonhardt, Ainsley Bristol, Darren G. Candow, Philip D. Chilibeck

**Affiliations:** 1Faculty of Kinesiology and Recreation Management, University of Manitoba, Winnipeg, MB R3T 2N2, Canada; stephen.cornish@umanitoba.ca; 2Centre on Aging, University of Manitoba, Winnipeg, MB R3T 2N2, Canada; forbess@brandonu.ca; 3Faculty of Graduate Studies, Applied Health Sciences, University of Manitoba, Winnipeg, MB R3T 2N2, Canada; umcordid@myumanitoba.ca; 4Pan Am Clinic Foundation, Winnipeg, MB R3M 3E4, Canada; 5College of Kinesiology, University of Saskatchewan, Saskatoon, SK S7N 5B2, Canada; keely.shaw@usask.ca (K.A.S.), tpl314@mail.usask.ca (T.L.), aib690@mail.usask.ca (A.B.); 6Department of Physical Education Studies, Faculty of Education, Brandon University, Brandon, MB R7A 6A9, Canada; 7Faculty of Kinesiology and Health Studies, University of Regina, Regina, SK S4S 0A2, Canada; darren.candow@uregina.ca

**Keywords:** sarcopenia, PUFA, strength, inflammation, elderly, anabolism

## Abstract

Sarcopenia negatively affects skeletal muscle mass and function in older adults. Omega-3 (ω-3) fatty acid supplementation, with or without resistance exercise training (RET), is suggested to play a role as a therapeutic component to prevent or treat the negative effects of sarcopenia. A systematic review and meta-analysis were conducted on the impact of ω-3 fatty acid supplementation with or without RET on measures of muscle mass and function in older adults (≥55 y). The data sources included SPORTDiscus, PubMed, and Medline. All the study types involving ω-3 fatty acid supplementation on measures of muscle mass and function in older adults (without disease) were included. The mean differences (MDs) or standardized mean differences (SMDs) with 95% confidence intervals were calculated and pooled effects assessed. Sixteen studies (1660 females, 778 males) met our inclusion criteria and were included in the meta-analysis. ω-3 fatty acid supplementation did not impact lean tissue mass (SMD 0.09 [−0.10, 0.28]). Benefits were observed for lower body strength (SMD 0.54 [0.33, 0.75]), timed-up-and-go (MD 0.29 [0.23, 0.35]s), and 30-s sit-to-stand performance (MD 1.93 [1.59, 2.26] repetitions) but not walking performance (SMD −0.01 [−0.10, 0.07]) or upper body strength (SMD 0.05 [−0.04, 0.13]). Supplementing with ω-3 fatty acids may improve the lower-body strength and functionality in older adults.

## 1. Introduction

Sarcopenia, characterized by the age-related reduction in strength and muscle mass, is a global health issue [1]. A recent systematic review and meta-analysis found that sarcopenic older adults had reduced functionality and were at greater risk of experiencing a fall compared to non-sarcopenic adults [2]. The direct healthcare costs related to treating sarcopenia in the United States of America were $18.5 billion in 2000 [3], which is projected to increase to $30.5 billion in 2022 (inflation estimated with https://www.in2013dollars.com, accessed on 10 April 2022). This cost has undoubtedly risen even more due to the aging population, and it is suggested that more high-quality research utilizing cost-effective strategies to counteract these effects is warranted [3]. Further, several scientific studies have demonstrated the association of sarcopenia with metabolic function and inflammation [4,5,6,7,8]. Chronic low-grade inflammation associated with aging (i.e., ‘inflammaging’) is known to precipitate metabolic syndrome, which may have a reciprocal function in perpetuating the ‘inflammaging’, creating a vicious cycle leading to skeletal muscle mass loss due to the catabolic activity of pro-inflammatory mediators [9].

Resistance exercise training (RET) could be considered standard care for older adults at risk of sarcopenia and also has a variety of other health benefits (for recent reviews, see: Refs. [10,11]). When completed as part of a healthy lifestyle in older adults, RET enhances skeletal muscle mass and strength [10,12,13]. Additionally, RET is anti-inflammatory, reducing ‘inflammaging’ and enhancing health outcomes in older adults [14]. Thus, RET could be considered standard care for older adults at risk of various health conditions, especially sarcopenia [15,16].

Supplementing habitual dietary intake with omega-3 (ω-3) fatty acids has recently gained attention for its potential anti-sarcopenic effects [17]. A typical Western diet tends to have a high ratio of ω-6 fatty acids to ω-3 fatty acids, which may contribute to the inflammatory milieu with advancing age and muscle loss [18]. Linoleic acid, the parent compound for ω-6, is converted to arachidonic acid, a precursor to eicosanoids (which include prostaglandins, prostacyclins, thromboxanes, and leukotrienes), which promote the production of pro-inflammatory cytokines [19]. Such an increase in inflammation could perpetuate the loss of muscle mass and strength associated with increasing age. On the other hand, the parent compound for ω-3 fatty acids, alpha-linolenic acid (ALA), is metabolized to form the poly-unsaturated fatty acids eicosapentaenoic acid (EPA) and docosahexaenoic acid (DHA). These ω-3 fatty acids are precursors to the formation of eicosanoids, which are less inflammatory or are anti-inflammatory compared to those produced by ω-6 fatty acids. ω-3 fatty acids compete with the enzymes cyclooxygenase and lipoxygenase involved in eicosanoids formation from arachidonic acid [19,20]. This competitive interference with arachidonic acid could result in ω-3 fatty acids reducing inflammation [21].

A number of systematic reviews on the effect of ω-3 fatty acid supplementation on muscle in older adults have recently been published [22,23,24]; however, they are limited in that they included older adults with muscle wasting due to diseased states (such as cancer) or did not compare supplementation during RET versus supplementation without RET. Further, this area of research has been expanding rapidly and, therefore, an update to the current state of the literature is warranted. Our purpose was to conduct an updated systematic review and meta-analysis investigating the effects of supplementing ω-3 fatty acids on measures of muscle mass and muscular performance, either alone or in combination with RET, in older adults. We hypothesized that ω-3 fatty acid supplementation would be effective for improving the lean tissue mass (i.e., muscle mass) in older adults and this would translate into greater strength and functional performance.

## 2. Materials and Methods

This systematic review was completed in accordance with the Preferred Reporting Items for Systematic Review and Meta-Analysis (PRISMA) statement [25]. A literature search was conducted using PubMed, SPORTDiscus, and MedLine, including all dates from inception up to 7 May 2022. The following Boolean phrase was used: (omega-3 OR omega 3 OR n-3 OR fishoil OR fish oil OR EPA OR DHA OR ALA or eicosapentaenoic acid OR docosahexaenoic acid OR acid α-linolenic acid OR alpha- linolenic acid) AND (resistance training OR muscle function OR hypertrophy OR atrophy) AND (older adult OR elderly OR aging OR sarcopenia). No restriction on language or date was implemented. Articles meeting the following participants, intervention, comparators, outcomes, study type (PICOS) were included: Participants were adults ≥55 years old, without major disease conditions (for example, cancer, chronic obstructive pulmonary disease). The intervention of interest was supplemental fish oil or ω-3 (ALA, DHA, EPA) supplementation, with or without RET. Studies involving a multi-ingredient supplement (e.g., ω-3 plus whey protein or other nutritional supplements) were not included. Studies involving a control (placebo) substance or no supplement as a comparator were included. Outcomes of interest included lean tissue (i.e., muscle) mass, lower- and upper body strength, and measures of functional ability (i.e., walking performance, timed-up-and-go test, repeated sit-to-stand testing). Randomized controlled trials as well as prospective study types were included. Only published material was included. Reference lists of included articles were also searched for any other articles that may fit our inclusionary criteria based on the above PICOS parameters.

Titles, abstracts, and full texts were screened to determine eligibility by two researchers, with any conflict settled by a third researcher. Two researchers completed the risk of bias for each article using the revised Cochrane risk of bias tool [26] and a third researcher settled disagreements. Sensitivity analyses were performed by removing studies with a high risk of bias or publication bias (i.e., from funnel plots) to determine if they impacted the outcomes of the meta-analyses.

Data were extracted from included manuscripts as means and standard deviations for intervention and comparator groups. Meta-analyses were run using RevMan 5.3 software (Cochrane Community, London, UK) using fixed-effects models. Mean differences (MDs) and 95% confidence intervals (CIs) or standardized mean differences (SMDs; where measurement techniques on the same outcome varied between studies) between intervention and control groups were calculated. Heterogeneity was assessed using χ^2^ and I^2^ tests, where heterogeneity was indicated by χ^2^
*p*-values ≤ 0.1 or I^2^ test value > 75%. When sufficient data were available, forest plots were generated using pooled effects. All data were assessed using standard mean difference, with the exception of the timed-up-and-go (TUG) and sit-to-stand tests, both of which were assessed using mean difference as the tests were the same across studies for these two measures. Funnel plots were generated to determine whether there was publication bias.

## 3. Results

From our search strategy, 1885 articles resulted, with 1859 remaining after duplicates were removed (Figure 1). After screening by title and abstract, 22 full text articles were reviewed for inclusion, of which 14 articles met our inclusion criteria for our review. Two further articles were identified for inclusion from reference lists of other articles, resulting in 16 articles being included in our review (Table 1). These articles included 2438 participants (1660 females, 778 males). Fifteen studies included were randomized controlled trials [27,28,29,30,31,32,33,34,35,36,37,38,39,40,41], while one was a randomized, non-controlled trial [42].

### 3.1. Participant Characteristics

The average age of the participants was 69.1 years. Seven studies involved females only [29,31,32,34,35,41,42], one study involved males only [37], while the remainder included both males and females [27,28,30,33,36,38,39,40]. Nine studies involved untrained individuals [27,28,29,33,37,38,40,41,42], one involved recreationally active participants [35], and six did not report the activity levels of the participants [30,31,32,34,36,39]. Rolland et al. [30] involved individuals at risk of cognitive decline, Félix-Soriano et al. [31] involved overweight or obese individuals, Krzymińska-Siemaszk et al. [39] included those with low lean body mass or at risk of having low lean body mass, twelve studies included individuals who were free of any significant health concerns [28,29,32,33,34,35,36,37,38,40,41,42], and one did not report the health status of the participants [27]. Eight studies involved a structured RET program [27,31,34,35,36,37,38,42], two involved a mixed program involving circuit training, postural exercises, and aerobic exercise [29,32], five did not involve an exercise program [33,34,39,40,41], and one involved counseling to encourage the participants to reach 150 min of physical activity per week [30].

### 3.2. Intervention and Comparators

The supplementation regimes varied vastly among the studies. Details on the doses used are outlined in Table 1. The majority of the studies used fish oil or fish-oil-derived ω-3 fatty acids [28,31,33,34,38,41,42], one used krill [40], one used flax [27], Da’ová et al. [29] and Štěpán et al. [32] used Calanus oil, and the remainder did not report the source of ω-3 [30,35,36,37,39]. For comparators, four studies used corn oil [27,33,35,36], two used sunflower oil [29,32], two used olive oil [34,39,41], two used safflower oil [28,38], and one each used a mixed vegetable oil [40], a blend of ω-3-6-9 fatty acids [37], vitamin E [39], and paraffin oil [30], respectively, and one did not use a control substance [42].

### 3.3. Risk of Bias

Of the sixteen articles included, three had a high risk of bias [34,39,42], four had some concerns [27,28,37,41], and nine had a low risk of bias (Table 2).

Studies with a high risk of bias were rated as such in the domain of risk of bias in the “measurement of the outcome” due to the lack of blinding of investigators, which may impact the assessment of factors such as strength. Studies with some concerns were rated as such in the “selection of reported results” because it could not be determined whether the trial was analyzed according to a pre-specified plan (i.e., the study was not registered before recruitment).

### 3.4. Lean Tissue Mass

A forest plot of the effect of ω-3 fatty acids on lean tissue mass is shown in Figure 2. Five articles reported the effects of ω-3 fatty acid supplementation alone on lean tissue mass in older adults [31,32,34,39,40], involving 297 participants (*n* = 58 male, *n* = 239 female). No significant difference was observed between the ω-3 fatty acids and control (*p* = 0.50). Five articles investigate the impact of ω-3 fatty acids in addition to RET in older adults [28,30,32,36,38], involving 217 participants (52 males, 165 females). Cornish & Chilibeck [27] reported the male and female results separately; therefore, these were assessed individually in the forest plot (Figure 2). No impact of ω-3 fatty acid supplementation in conjunction with RET was found for lean tissue mass (*p* = 0.48). No significant difference was observed when all the studies were taken together (with and without RET) (*p* = 0.33).

### 3.5. Lower Body Strength

Pooled effects of ω-3 fatty acid supplementation on lower body strength with and without RET are displayed in Figure 3. Three articles [31,33,40] (209 participants; *n* = 56 males, 153 females) investigated the effects of ω-3 fatty acid supplementation on lower body strength without RET. The pooled analysis indicated a significant positive impact of ω-3 without RET on lower body strength (*p* < 0.001). Seven articles reported the effects of ω-3 fatty acid supplementation in conjunction with RET on lower body strength [27,31,35,36,37,38,42] (*n* = 288; 102 males, 186 females). Both Cornish & Chilibeck [27] and Da Boit et al. [38] reported male and female results independently and, thus, they are displayed as such in the pooled analysis (Figure 3). Leg strength was significantly improved in the ω-3 group compared to the control (*p* = 0.005), although significant heterogeneity was present (I^2^ = 86% *p* < 0.001). When the study with a high risk of bias [42] was removed, the significant finding was no longer present (SMD 0.22 (−0.08,0.51), *p* = 0.15). When considering all the studies (with and without RET) after the high-risk study was removed, there was still a significant effect favoring the ω-3 over the control for lower body strength (*p* < 0.001), and a subgroup difference was present (*p* = 0.03), with ω-3 supplementation being more effective in increasing lower body strength when taken on its own compared to when taken with RET.

Two studies were identified to have publication bias from funnel plot analysis (i.e., they fell outside the funnel) [38,42]. When these were removed, the violation of heterogeneity was no longer present. With these removed, the findings for ω-3 combined with RET were not significant, but the overall findings of ω-3 supplementation (with and without RET) were still significant (*p* = 0.0007), and a significant subgroup difference was present, with ω-3 supplementation being superior when taken on its own when compared to taken in conjunction with RET (*p* = 0.005).

The pooled effects of ω-3 fatty acid supplementation on upper body strength with and without RET are displayed in Figure 4. When considered without RET, seven articles were analyzed [30,31,33,34,39,40,41], involving 2083 participants (*n* = 666 males, 1417 females). No significant effects of ω-3 were observed (*p* = 0.11), although significant heterogeneity was observed (I^2^ = 81%; *p* < 0.001). When studies with a high risk of bias or publication bias [29,31,34,39] were removed, there was a significant effect on upper body strength when ω-3 was supplemented without RET (SMD 0.11 [0, 0.23]) (*p* = 0.05). Heterogeneity was reduced but still significant (I^2^ = 77%; *p* = 0.002). Removing one of the remaining studies at a time still did not reduce the heterogeneity to a non-significant level. Seven articles reported ω-3 supplementation in conjunction with RET on upper body strength [27,28,29,30,31,36,37] (*n* = 1931; *n* = 670 males, 1261 females). No significant findings were present (*p* = 0.88), but there was significant heterogeneity (I^2^ = 81%; *p* < 0.001). Removing studies with publication bias [29,31,34,39] reduced the heterogeneity (I^2^ = 0%), but this did not affect the level of significance (SMD 0.02 [−0.10, 0.15] (*p* = 0.73). There was no subgroup difference whether or not studies with a high risk of bias or publication bias were removed.

### 3.6. Functional Performance Tests

Six studies reported TUG results [28,34,36,37,39,42] (*n* = 197; 66 males, 131 females). Of these, only two did not include an RET intervention [34,39]; we, therefore, did not include a sub-group analysis to compare studies with and without RET. The pooled analyses indicated improved performance in TUG (i.e., reduced time) with ω-3 (*p* < 0.001) (Figure 5). These findings remained even when studies with a high risk of bias [34,39] were removed.

Six studies evaluated 30-s sit-to-stand performance [28,29,32,34,36,42] (*n* = 230; 25 males, 205 females). Only a single study did not involve RET; therefore, we did not perform sub-group analyses. The pooled analyses (Figure 6) indicated a significant effect favoring ω-3 (*p* < 0.001), although there was significant heterogeneity (I^2^ = 92%; *p* < 0.001). When studies with a high risk of bias (which also had publication bias) [34,42] were removed, the heterogeneity was eliminated (I^2^ = 0%), but the effect of ω-3 was no longer significant (MD 0.58 [−0.12, 1.28] repetitions) (*p* = 0.11).

Four studies [30,39,40,41] investigated walking speed without a RET intervention (*n* = 1944; 651 males, 1293 females). No difference was observed between the conditions (Figure 7). When the study with a high risk of bias and publication bias [39] was removed, the results remained non-significant. Six articles assessed the effect of ω-3 supplementation in addition to RET on walking speed [28,30,36,37,38,42] (Figure 7; *n* = 1833; 653 males, 1180 females). There was no significant effect of ω-3 supplementation (Figure 7; *p* = 0.54). These results were unchanged when articles with a high risk of bias [39,42] or publication bias [41] were removed. When combining articles with and without RET, no significant effect of ω-3 supplementation was observed (*p* = 0.79).

## 4. Discussion

The most important finding of this systematic review and meta-analysis is that ω-3 supplementation does not impact lean tissue mass with or without RET in older adults but does improve lower body strength and lower body functional performance. Supplementation was marginally effective in studies without resistance training for improving upper body strength, but only after studies high in risk of bias or publication bias were removed from analysis. Supplementation appears to be effective when taken alone, but less so when taken in conjunction with RET for improving strength. No impacts were observed for walking speed. Our finding that ω-3 supplementation improves lower body strength and functional ability is important because lower body strength is preferentially affected by aging [43], and sarcopenic older adults suffer from reduced functional performance [2]. Tests such as the TUG and sit-to-stand test, which were improved with ω-3 supplementation in our meta-analyses, are important predictors of functional ability in older adults [44,45,46].

Our hypothesis that ω-3 supplementation would enhance lean tissue mass (i.e., muscle mass) was based on a variety of narrative reviews, which have speculated that ω-3 supplementation may enhance skeletal muscle anabolism [47,48,49]. There are several physiological mechanisms by which ω-3 supplementation might improve muscle mass: ω-3 supplementation may reduce inflammation [24], which might reduce the activation of pathways involved in protein degradation [21]. ω-3 supplementation activates signaling proteins involved in the activation of translation [32,50] and enhances protein synthesis in response to amino acids or insulin [50]. Other studies have failed to find evidence for ω-3 supplementation enhancement of protein synthesis; for example, ω-3 supplementation during resistance training does not lead to increased activation of satellite cells or incorporation of satellite cells into muscle fibres as new myonuclei [32].

An enhancement of muscular strength with ω-3 supplementation, as found in our meta-analysis, might occur in the absence of improvements in muscle mass if neural activation is improved. ω-3 fatty acids may be incorporated into nervous tissue cell membranes [51], and this may affect the neural activation of muscle. Further, ω-3 supplementation during resistance training programs in older adults improves the ability to recruit motor units and reduces electromechanical delay, indicating enhanced neural activation of muscle [39]. Changes such as reduced electromechanical delay (i.e., a reduction in the time from neural activation to production of muscular force) could explain some of the improvements in functional ability (i.e., the TUG test, number of sit-to-stand repetitions within 30 s) observed with ω-3 supplementation in our meta-analyses.

Given the irrefutable role of RET in the delay and attenuation of sarcopenia [52,53,54], RET should always be the cornerstone of both preventative and treatment plans for sarcopenia. However, with less than half of older adults meeting physical activity guidelines [55], other strategies to mitigate the effects of aging on muscle are of interest. A variety of nutritional supplements, such as creatine monohydrate, whey protein, and vitamin D, have been used in older adults to combat the development and progression of sarcopenia [21]. The results of the current meta-analysis indicate that ω-3 supplementation improves lower body strength, but to a lesser amount when combined with resistance training. The very high stimulus of resistance training for improving strength may mask the effects of a dietary supplement.

There are several limitations with the findings from our meta-analyses. Some of our results were affected by the removal of studies deemed to have a high risk of bias or publication bias. These studies contributed to heterogeneity across our meta-analyses. When studies with a high risk of bias or publication bias were removed, ω-3 supplementation was no longer effective for improving lower body strength when taken during resistance training programs. Likewise, when studies with a high risk of bias were removed, the significant impact of ω-3 supplementation for improving sit-to-stand performance was lost. On the other hand, when studies with a high risk of bias or publication bias were removed from the upper body strength meta-analysis, ω-3 supplementation was marginally effective for improving strength in studies without a resistance-training component. These findings indicate a need for clinical trials that have a lower risk of bias.

## 5. Conclusions

The current systematic review and meta-analysis outline the impact of ω-3 supplementation on muscular mass and function in older adults. Supplementation with ω-3 fatty acids may improve strength and functional ability in older adults but does not appear to influence lean body mass. Overall, more research on ω-3 fatty acid supplementation in skeletal muscle is a high priority considering the potential benefits these types of fatty acids may have in reducing sarcopenia and dynapenia. It is recommended that high-quality randomized controlled trials with larger sample sizes are needed to fully elucidate the effects of ω-3 fatty acid supplementation on skeletal muscle mass and skeletal muscle function in older adults at risk of sarcopenia. Further, future research should elucidate any differences in the source of ω-3 fatty acids, such as plant-based (flax), plankton, krill, and fish. Optimal dosing strategies for improving muscle mass and function are also an area for future research.

## Figures and Tables

**Figure 1 nutrients-14-02221-f001:**
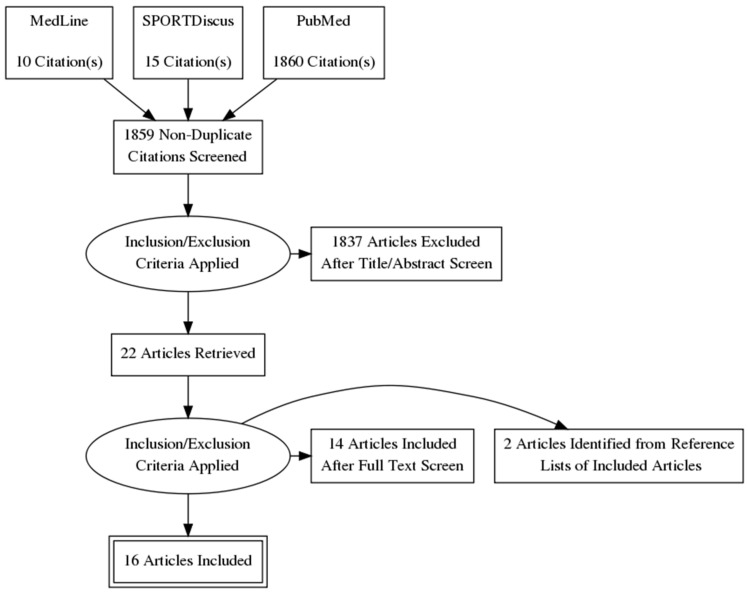
PRISMA diagram: flow chart of study selection process.

**Figure 2 nutrients-14-02221-f002:**
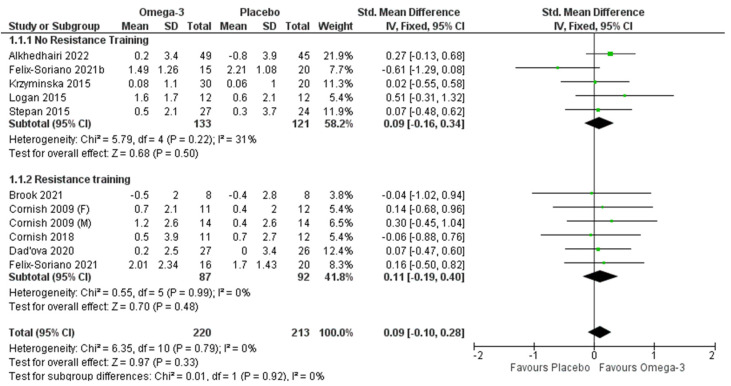
Pooled analysis of the impact of omega-3 supplementation on lean body mass with and without resistance training [27,29,31,34,35,37,39,40].

**Figure 3 nutrients-14-02221-f003:**
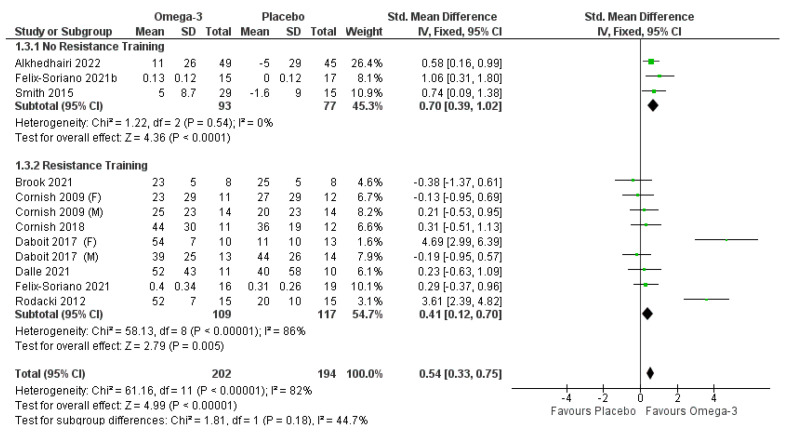
Pooled analysis of the impact of omega-3 supplementation on lower body strength with and without resistance training [27,31,33,35,36,37,40,42].

**Figure 4 nutrients-14-02221-f004:**
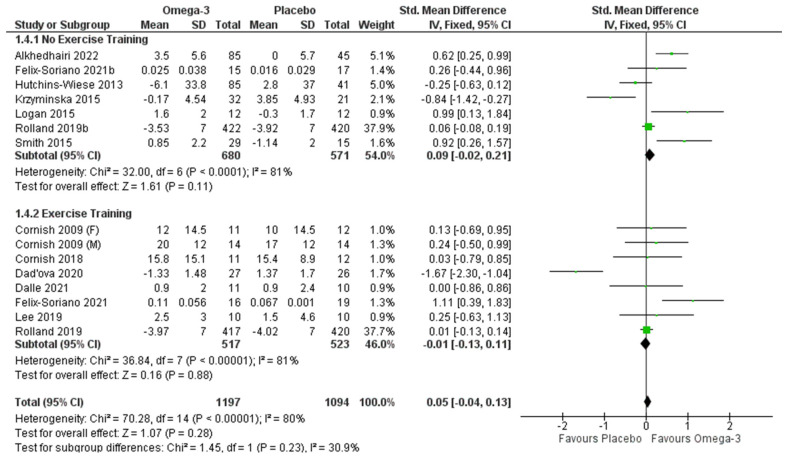
Pooled analysis of the impact of omega-3 supplementation with and without resistance training on upper body strength [27,28,29,30,31,33,34,36,37,39,40,41].

**Figure 5 nutrients-14-02221-f005:**
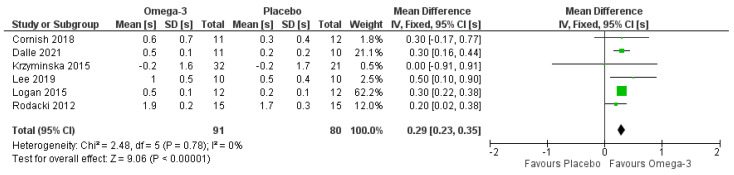
Pooled analysis of the impact of omega-3 supplementation on timed-up-and-go test. Note that all mean results entered for each study are reduction in seconds within groups [28,34,36,37,39,42].

**Figure 6 nutrients-14-02221-f006:**
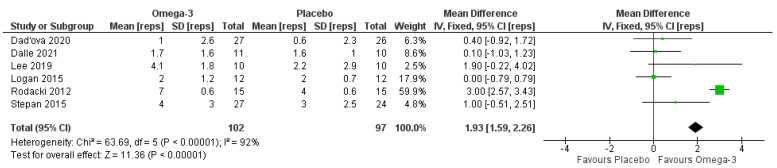
Pooled analysis of the impact of omega-3 supplementation on sit-to-stand performance.

**Figure 7 nutrients-14-02221-f007:**
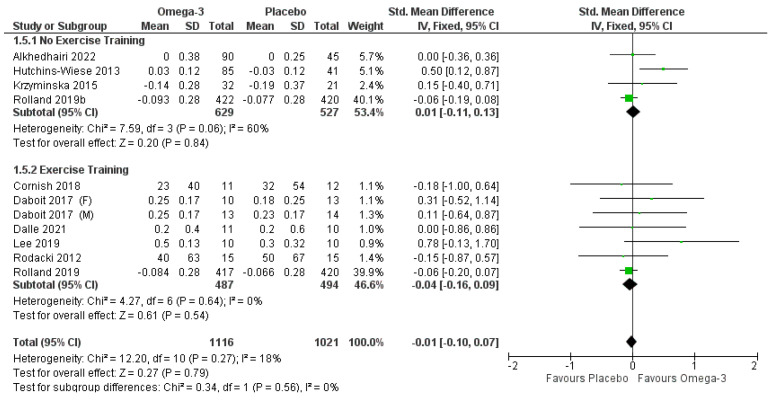
Pooled analysis of the impact of omega-3 supplementation on walking speed.

**Table 1 nutrients-14-02221-t001:** Studies investigating the effects of omega-3 supplementation on measures of muscular health in older adults.

Author	Design	Sample	Intervention	Main Results
No RET				
Alkhedhairi et al. [40]	Double-blind RCT	*n* = 94; female, *n* = 53; male, *n* = 41; age = 71.2 ± 5.1 years	Krill oil (4 g/day; 772 mg/d EPA and 384 mg/day DHA) or placebo (4 g/day mixed vegetable oil) for 6 months	Krill oil supplementation resulted in improved knee extensor maximal torque (9.3%), grip strength (10.9%), and vastus lateralis muscle thickness (3.5%) to a greater extent than placebo. However, there was no difference in short performance physical battery test between groups.
Hutchins-Wiese et al. [41]	Double-blind RCT	*n* = 126; all female; age, 75 (range 64–95) years	ω-3 (1.2 g/day EPA and 1.2 g/day DHA) or placebo (1.8 g/day olive oil)	Higher RBC DHA content and DHA/AA ratio was associated with less frailty (*p* = 0.007 and *p* = 0.004, respectively). Fish oil supplementation improved walking speed compared to placebo (*p* = 0.038).
Krzymińska-Siemaszko et al. [39]	Non-blinded RCT	*n* = 50; 17 male and 33 female; age, 74.6 ± 8.0 years; all with decreased muscle mass	ω-3 (1.3 g/day PUFA (2 capsules/day containing 600 mg EPA, 440 DHA, 200 mg other ω-3 fatty acids) and 10 mg/day vitamin E) or placebo (11 mg/day vitamin E solution) for 12 weeks	No difference in muscle mass, grip strength, timed-up-and-go test, or appendicular lean mass index.
Logan and Spriet [34]	Single-blind RCT	*n* = 24; all female; age = 66 ± 1 years	ω-3 (5 g/day (2 g/day EPA and 1 g/day DHA)) or placebo (3 g/day olive oil) for 12 weeks	ω-3 supplementation resulted in increased lean mass (4%) and timed-up-and-go test (7%), while no improvements were observed in the placebo group. No improvements were observed in grip strength or 30-s sit-to-stand test for either group.
Rolland et al. [30]	Double-blind RCT	*n* = 1679; age = 75.34 ± 4.42 years	ω-3 (800 mg/day DHA and 225 mg/day EPA), placebo (paraffin oil), ω-3 and a multidomain intervention (including physical activity and nutrition advice, and cognitive training), or placebo and multidomain intervention for 36 months	No differences between groups were found for chair-stand test, handgrip strength, 4-m walking speed, or short physical performance battery.
Smith et al. [33]	Double-blind RCT	*n* = 44; male = 15 and female = 29; age, control = 69.7 ± 7 years and omega-3 = 68 ± 5 years (mean ± SEM)	ω-3 (4 × 1 g pills/day providing 1.86 g/day EPA and 1.5 g/day DHA) or placebo (4 × 1 g pills/day of corn oil) for 6 months	ω-3 supplementation increased thigh muscle volume, handgrip strength, and 1 repetition maximum muscle strength compared to control. Average isokinetic power approached significance with ω-3 supplementation as well (*p* = 0.075).
With RET				
Brook et al. [35]	Double-blind RCT	*n* = 16; female; age, placebo = 66.5 ± 1.4 years and ω-3 = 64.4 ± 0.8 years	ω-3 PUFA (3680 mg/day (1860 mg EPA and 1540 mg DHA)) or placebo (corn oil) for 6 weeks	1 repetition maximum and number of myonuclei in type I and type II fibres increased equally in treatment arms. ω-3 supplementation resulted in greater thigh fat free mass and type II fibre cross sectional area, as well as greater 4EBP1 activation after acute RE at the 6-week time-point compared to placebo. No differences in maximum voluntary contraction, type I fibre cross sectional area, and satellite cell number were observed between groups.
Cornish and Chilibeck [27]	Double-blind RCT	*n* = 51; age, 65.4 ± 0.8 years; male = 28, female = 23	Flaxseed oil 30 mL/day (~14 g/day ALA) or placebo (30 mL/day corn oil) for 12 weeks	Males supplementing with ALA demonstrated decreased systemic IL-6 concentrations and increased knee flexor muscle thickness following 12 weeks of RET. Females demonstrated no additional benefit associated with ALA supplementation.
Cornish et al. [37]	Pilot double-blind RCT	*n* = 23; all male; age, ω-3 = 71.4 ± 6.2 years and placebo = 70.9 ± 5.0 years	3.0 g/day ω-3 (1.98 g EPA and 0.99 g DHA) or placebo (ω 3-6-9 blend, 1350 mg ALA, 795 mg linoleic acid and γ-linolenic acid, 525 mg oleic acid, 330 mg of other short-chain fatty acids, saturated fat, and phospholipids) for 12 weeks	RET improved lean tissue mass, chest press and leg press strength, and physical function, with no added benefits with ω-3 supplementation.
Da Boit et al. [38]	Double-blind RCT	*n* = 50; male: *n* = 27 and female: *n* = 23; age, male = 70.6 ± 4.5 years and female = 70.7 ± 3.3 years	ω-3 (3.0 g/day fish oil) or placebo (3 g/day safflower oil) for 18 weeks	In females supplemented with ω-3 fatty acids, maximal isometric torque and muscle quality improved to a greater extent than placebo, with no difference in males.
Daďová et al. [29]	Double-blind RCT	*n* = 55; all female; age = 70.9 ± 3.9 years	Calanus oil (~105 mg/day DHA and 125 mg/day EPA) or placebo (sunflower oil) and combined aerobic and RET training for 16 weeks	Calanus oil improved chair-stand test repetitions (calanus oil, median ∆ = 4 vs. placebo median ∆ = 3) but not muscle mass compared to placebo.
Dalle et al. [36]	Double-blind RCT	*n* = 23; male: *n* = 8 and female: *n* = 15; age range, 65–84 years	ω-3 (1100 mg three times/day (410 mg DHA, 540 mg EPA and 4 mg vitamin E) or placebo (1100 mg corn oil three times/day) for 14 weeks	ω-3 supplementation enhanced isometric strength gains but not muscle volume, catabolic, or inflammatory adaptations in response to RET.
Félix-Soriano et al. [31]	Double-blind RCT	*n* = 67; all overweight/obese females; age range, 55–70 years	Placebo (3 g/day olive oil), ω-3 (3 g/day containing 1650 mg DHA and 150 mg EPA), placebo and RET, omega-3 and RET for 16 weeks	RET resulted in improved upper limb lean mass, muscle strength, and muscle quality compared to the untrained groups. ω-3 supplementation improved muscle quality of the lower limbs.
Lee et al. [28]	RCT	*n* = 28; 10 males and 18 females; age, 66.5 ± 5.0 years	ω-3 (2.1 g/day EPA and 0.72 g/day EHA) and RT, placebo (safflower oil) and RET, and control only for 12 weeks	RET resulted in improved handgrip strength, five times sit-to-stand, timed-up-and-go, 6-m walk, and 30-s sit-to-stand.
Rodacki et al. [42]	Randomized, non-controlled	*n* = 45; all female; age = 64 ± 1.4 years	All completed 90 RET. One group only did RET; another consumed fish oil during RET; lastly, a group consumed fish oil for 60 days prior to RET. Participants receiving fish oil consumed 2 g/day (~0.4 g/d EPA and 0.3 g/day DHA)	Both groups that consumed fish oil had greater improvements in peak torque and rate of torque development and chair-rising performance compared to the RET only group.
Štěpán et al. [32]	Double-blind RCT	*n* = 55; all female; placebo, age = 70 ± 4 years; Calanus oil, age = 71 ± 4 years	Calanus oil (~230 mg/day EPA + DHA) or placebo (sunflower oil) combined aerobic and RET for 4 months	Exercise training resulted in improved lean body mass, arm curl repetitions, and chair-stand test for both groups. Additionally, an interaction effect was identified for chair-stand test (mean ∆, calanus oil = 4 vs. placebo = 3), indicating calamus oil supplementation may contribute to greater improvements.

RBC, red blood cell; PUFAs, polyunsaturated fatty acids; AA, arachidonic acid; EPA, eicosapentaenoic acid; ALA, a-linolenic acid; DHA, docosahexaenoic acid; ω-3, omega-3 fatty acids; RET, resistance exercise training.

**Table 2 nutrients-14-02221-t002:** Risk of bias assessment for included studies.

Study	Risk of Bias Domain						
	Randomization Process	Period or Carry-Over Effect	Deviation from Intended Intervention	Missing Outcome Data	Measurement of Outcome	Selection of Reported Results	Overall Risk of Bias
Alkhedhairi et al. [40]	Low	Low	Low	Low	Low	Low	Low
Hutchins-Wiese et al. [41]	Low	Low	Low	Low	Low	Low	Low
Krzymińska-Siemaszko et al. [39]	Low	Low	Low	Low	High	Some Concerns	High
Logan and Spriet [34]	Low	Low	Low	Low	High	Some Concerns	High
Rolland et al. [30]	Low	Low	Low	Low	Low	Low	Low
Smith et al. [33]	Low	Low	Low	Low	Low	Low	Low
Brook et al. [35]	Low	Low	Low	Low	Low	Low	Low
Cornish and Chilibeck [27]	Low	Low	Low	Low	Low	Some Concerns	Some Concerns
Cornish et al. [37]	Low	Low	Low	Low	Low	Some Concerns	Some Concerns
Da Boit et al. [38]	Low	Low	Low	Low	Low	Low	Low
Daďová et al. [29]	Low	Low	Low	Low	Low	Low	Low
Dalle et al. [36]	Low	Low	Low	Low	Low	Low	Low
Félix-Soriano et al. [31]	Low	Low	Low	Low	Low	Low	Low
Lee et al. [28]	Low	Low	Low	Low	Low	Some Concerns	Some Concerns
Rodacki et al. [42]	Low	Low	Low	Low	High	Some Concerns	High
Štěpán et al. [32]	Low	Low	Low	Low	Low	Low	Low

## Data Availability

Data are available upon request.

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
