# Peer review of "Effects of Omega-3 Supplementation Alone and Combined with Resistance Exercise on Skeletal Muscle in Older Adults: A Systematic Review and Meta-Analysis"

_nutrients, 2022, doi:10.3390/nu14112221_

Round 1
Reviewer 1 Report
The manuscript by Cornish et al. describes the benefits of Omega-3 supplementation alone and combined with resistance exercise on skeletal muscle in adults and its implications in managing muscular diseases like Sarcopenia. This is an exciting topic and a well-written manuscript.
Besides that, I have some major concerns about the novelty of the concept and the study design. My suggestions/comments are as follows:
- Although the topic is interesting, it lacks novelty. There are plenty of narrative reviews on similar/related topics. The list below can be referred to for similar studies.
https://journals.lww.com/co-clinicalnutrition/Abstract/2014/03000/Omega_3_fatty_acids_and_protein_metabolism_.7.aspx
https://www.mdpi.com/2072-6643/12/7/2057
https://physoc.onlinelibrary.wiley.com/doi/full/10.1113/JP275443
https://www.frontiersin.org/articles/10.3389/fnut.2019.00144/full
https://link.springer.com/article/10.1007/s40520-019-01146-1
doi: 10.1016/j.maturitas.2019.11.007
doi: 10.1017/S0029665119000922.
doi: 10.1007/s40520-019-01146-
doi: 10.1097/MCO.0000000000000441.
- The best way to enhance its novelty and make it impactful is to convert it to scoping or systematic review, which will make it more comprehensive and exciting for the readers, from clinicians to basic science researchers.
- The systematic literature review will provide the most updated and complete literature related to the keywords (i.e., Omega-3 Supplementation; Resistance Exercise; Muscle; sarcopenia).
- Systematic data extraction and quantitative/qualitative representation of data in the result section (such as tables 1 & 2 in the manuscript) will be more impactful.
- Omega-3 supplementation alone and combined with resistance exercise on- Primary outcomes (muscular function/mass/health) and Secondary outcomes (inflammatory markers and other molecular markers) could have been summarised in the tables and forest plots.
- Subsections- Omega-3 Fatty Acids Alone, and Omega-3 fatty acids and resistance exercise training (RET), could have presented as results, including tables 1 & 2.
For conducting the scoping review or systematic review, the following references could be followed:
Peters MDJ, Godfrey CM, Khalil H, et al. Guidance for conducting systematic scoping reviews. JBI Evid Implement 2015; 13: 141–146.
Munn Z, Peters MDJ, Stern C, et al. Systematic review or scoping review? Guidance for authors when choosing between a systematic or scoping review approach. BMC Med Res Methodol 2018; 18: 1–7.
https://tropmedhealth.biomedcentral.com/articles/10.1186/s41182-019-0165-6
Author Response
The manuscript by Cornish et al. describes the benefits of Omega-3 supplementation alone and combined with resistance exercise on skeletal muscle in adults and its implications in managing muscular diseases like Sarcopenia. This is an exciting topic and a well-written manuscript.
Besides that, I have some major concerns about the novelty of the concept and the study design. My suggestions/comments are as follows:
- Although the topic is interesting, it lacks novelty. There are plenty of narrative reviews on similar/related topics. The list below can be referred to for similar studies.
https://journals.lww.com/co-clinicalnutrition/Abstract/2014/03000/Omega_3_fatty_acids_and_protein_metabolism_.7.aspx
https://www.mdpi.com/2072-6643/12/7/2057
https://physoc.onlinelibrary.wiley.com/doi/full/10.1113/JP275443
https://www.frontiersin.org/articles/10.3389/fnut.2019.00144/full
https://link.springer.com/article/10.1007/s40520-019-01146-1
doi: 10.1016/j.maturitas.2019.11.007
doi: 10.1017/S0029665119000922.
doi: 10.1007/s40520-019-01146-
doi: 10.1097/MCO.0000000000000441.
- The best way to enhance its novelty and make it impactful is to convert it to scoping or systematic review, which will make it more comprehensive and exciting for the readers, from clinicians to basic science researchers.
Response: the article has been converted to a systematic review
- The systematic literature review will provide the most updated and complete literature related to the keywords (i.e., Omega-3 Supplementation; Resistance Exercise; Muscle; sarcopenia).
Response: recommended keywords were included in search terms
- Systematic data extraction and quantitative/qualitative representation of data in the result section (such as tables 1 & 2 in the manuscript) will be more impactful.
Response: this has been done as part of the conversion to a systematic review
- Omega-3 supplementation alone and combined with resistance exercise on- Primary outcomes (muscular function/mass/health) and Secondary outcomes (inflammatory markers and other molecular markers) could have been summarised in the tables and forest plots.
Response: forest plots have been included for lean tissue mass, lower- and upper-body strength, and functional performance measures. We have summarized other results in tables.
- Subsections- Omega-3 Fatty Acids Alone, and Omega-3 fatty acids and resistance exercise training (RET), could have presented as results, including tables 1 & 2.
Response: These have been reported as results as part of the conversion to a systematic review
Reviewer 2 Report
This is a narrative review to provide an overview of studies that have assessed either omega-3 (ω-3) fatty acids alone or in combination with resistance-exercise training (RET) in older adults at risk of sarcopenia. This is an interesting and relevant topic for researchers and clinicians in the field of geriatrics. Despite calling it a narrative review, the authors present the findings as a type of systematic review but without describing the search methods. I believe they made an interesting search although it is still possible to improve the exposure of scientific evidence to facilitate the understanding of readers. The following are general and specific comments.
GENERAL COMMENTS
- Very extensive introduction. I suggest 3 paragraphs. I recommend that the introduction contemplate the state of the art of the possible role of omega-3 in the muscle of the older person and point out its possible mechanisms.
- Carefully review whether the studies in Table 1 in fact do not include strength exercises.
- I believe that the topic "multi-ingredient supplements including ω-3 fatty acids" does not fit in this review, and may appear at most as limitations in the search for studies, given the involvement of a series of supplements with a well-established effect on skeletal muscle (e.g. whey and creatine). Therefore, I believe that the authors could replace this topic with one that addresses the mechanisms of action of omega-3 in skeletal muscle, strengthening the present review.
- I believe that there is a need for a robust discussion of important aspects such as the variation of supplementation administered, dosage, blood levels, or strategies of use to better direct readers to effective strategies.
- The authors' conclusion does not summarize the findings of the review.
- I suggest a double-check to refine the writing throughout the manuscript.
SPECIFIC COMMENTS
Title.
It is not possible to consider "effect" in the title as the authors did not perform a meta-analysis from an RCT or an RCT.
Introduction
Page 1/Introduction/Paragraph 1 adequately addresses sarcopenia as the central axis of older adult muscle and functional decline and its cost. Page 2/Introduction/Paragraph 2, I believe that the authors should better address the process of chronic inflammation common in this population (inflammaging), discussing possible fields of action of omega-3. Page 2/Introduction/Paragraph 3, address the current panorama of studies that investigated the role of omega-3 and justify the reason for the possible additive role of the combined action in addition to the well-established role of strength exercise in the improvement of skeletal muscle (mass, strength, function and inflammation).
Page 1. Line 46. Inflammaging is a kind of chronic low-grade inflammation. Please change to (e.g., inflammaging)
Page 1. Line 46-49. I suggest rewriting for clear understanding: "Inflammaging, (include a brief definition), relates to the state of chronic low-grade inflammation that can precipitate metabolic syndrome while suggests perpetuating the 'inflammaging', creating..."
Page 2, Line 55: For accurate information I suggest writing directly: "RET could be considered standard care for older adults at risk of sarcopenia"
Page 2 Line 63-67. The focus of the review is omega-3, so I suggest removing the information from line 63 to 67.
Page 2 Line 66. This sentence is meaningless here. What reference supports this statement?
Page 2, Line 76. Please remove figure 1 from the introduction. I do not think it appropriate to include figures in the introduction as its role is to provide an overview of the subject. I suggest covering it in a topic after the introduction, if pertinent.
Page 3, Line 80-82. Please include reference.
Page 3, Table 1. This study did not only use the older adults (mean age 68.8 SD15.7 years). Were the authors careful to evaluate only the older adults subgroup (if the study presents this data) or would it be appropriate not to consider it?
Page 4. Table1. How do the authors justify that only omega-3 supplementation increased muscle protein synthesis? Wouldn't this study (Lalia et al., 32) have used a combined strategy (supplementation + exercise)?
Page 4. Table 1. The study by Murphy et al., (33) uses omega-6 supplementation, outside the scope of the review. Additionally, I reiterate the previous comment about Lalia et al., (32)
Page 5. Table 1. (Ref. Smith et al., 25) Again, see the comments on Lalia et al., (32) and Murphy et al., (33) regarding muscle protein synthesis.
Page 7. Table 2. The study by Bell et al., (38) is not within the scope of this review since the possible effect of the multi-ingredient beverage cannot be attributed to omega-3, despite being present in the composition. Please double-check all studies in Table 2.
Author Response
This is a narrative review to provide an overview of studies that have assessed either omega-3 (ω-3) fatty acids alone or in combination with resistance-exercise training (RET) in older adults at risk of sarcopenia. This is an interesting and relevant topic for researchers and clinicians in the field of geriatrics. Despite calling it a narrative review, the authors present the findings as a type of systematic review but without describing the search methods. I believe they made an interesting search although it is still possible to improve the exposure of scientific evidence to facilitate the understanding of readers. The following are general and specific comments.
GENERAL COMMENTS
- Very extensive introduction. I suggest 3 paragraphs. I recommend that the introduction contemplate the state of the art of the possible role of omega-3 in the muscle of the older person and point out its possible mechanisms.
Response: The introduction has been edited to outline the process of sarcopenia as well as the potential role for omega-3 fatty acids in the therapeutic approach to it.
- Carefully review whether the studies in Table 1 in fact do not include strength exercises.
Response: Tables 1 and 2 have been combined and absence or presence of resistance exercise training highlighted
- I believe that the topic "multi-ingredient supplements including ω-3 fatty acids" does not fit in this review, and may appear at most as limitations in the search for studies, given the involvement of a series of supplements with a well-established effect on skeletal muscle (e.g. whey and creatine). Therefore, I believe that the authors could replace this topic with one that addresses
the mechanisms of action of omega-3 in skeletal muscle, strengthening the present review.
Response- this has been removed. We provide a discussion of potential mechanisms in the discussion section.
- I believe that there is a need for a robust discussion of important aspects such as the variation of supplementation administered, dosage, blood levels, or strategies of use to better direct readers to effective strategies.
Response: differences in studies are highlighted in our revised conclusion. Supplementation and doses are outlined in detail in our table describing our included studies.
- The authors' conclusion does not summarize the findings of the review.
Response: the conclusion has been edited to better summarize the findings
- I suggest a double-check to refine the writing throughout the manuscript.
Response: Writing has been refined throughout
SPECIFIC COMMENTS
Title.
It is not possible to consider "effect" in the title as the authors did not perform a meta-analysis from an RCT or an RCT.
Response: We have updated the review to be systematic, therefore able to determine effect
Introduction
Page 1/Introduction/Paragraph 1 adequately addresses sarcopenia as the central axis of older adult muscle and functional decline and its cost.
Page 2/Introduction/Paragraph 2, I believe that the authors should better address the process of chronic inflammation common in this population (inflammaging), discussing possible fields of action of omega-3.
Response: We have discussed the impacts of inflammation and mechanisms by which omega-3 might decrease inflammation
Page 2/Introduction/Paragraph 3, address the current panorama of studies that investigated the role of omega-3 and justify the reason for the possible additive role of the combined action in addition to the well-established role of strength exercise in the improvement of skeletal muscle (mass, strength, function and inflammation).
Response: We have addressed the effects of omega-3 fatty acids with and without resistance training by adding a series of meta-analyses where these are compared in sub-analyses.
Page 1. Line 46. Inflammaging is a kind of chronic low-grade inflammation. Please change to (e.g., inflammaging)
Response: line 49-51- Chronic low-grade inflammation (i.e., ‘inflammaging’) is known to precipitate metabolic syndrome
Page 1. Line 46-49. I suggest rewriting for clear understanding: "Inflammaging, (include a brief definition), relates to the state of chronic low-grade inflammation that can precipitate metabolic syndrome while suggests perpetuating the 'inflammaging', creating..."
Response: this has been clarified (lines 49-51)
Page 2, Line 55: For accurate information I suggest writing directly: "RET could be considered standard care for older adults at risk of sarcopenia"
Response: this has been clarified (lines 54-55)
Page 2 Line 63-67. The focus of the review is omega-3, so I suggest removing the information from line 63 to 67.
Response: This has been removed
Page 2 Line 66. This sentence is meaningless here. What reference supports this statement?
Response: This has been re-written
Page 2, Line 76. Please remove figure 1 from the introduction. I do not think it appropriate to include figures in the introduction as its role is to provide an overview of the subject. I suggest covering it in a topic after the introduction, if pertinent.
Response: figure 1 has been removed
Page 3, Line 80-82. Please include reference.
Response: this has been deleted as part of compressing the introduction
Page 3, Table 1. This study did not only use the older adults (mean age 68.8 SD15.7 years). Were the authors careful to evaluate only the older adults subgroup (if the study presents this data) or would it be appropriate not to consider it?
Response: tables have been edited to include only those included in the meta-analysis
Page 4. Table1. How do the authors justify that only omega-3 supplementation increased muscle protein synthesis? Wouldn't this study (Lalia et al., 32) have used a combined strategy (supplementation + exercise)?
Response: tables have been edited to include only those included in the meta-analysis; therefore, this study has been removed
Page 4. Table 1. The study by Murphy et al., (33) uses omega-6 supplementation, outside the scope of the review. Additionally, I reiterate the previous comment about Lalia et al., (32)
Response: tables have been edited to include only those included in the meta-analysis
Page 5. Table 1. (Ref. Smith et al., 25) Again, see the comments on Lalia et al., (32) and Murphy et al., (33) regarding muscle protein synthesis. This study has been removed.
Response: tables have been edited to include only those included in the meta-analysis
Page 7. Table 2. The study by Bell et al., (38) is not within the scope of this review since the possible effect of the multi-ingredient beverage cannot be attributed to omega-3, despite being present in the composition. Please double-check all studies in Table 2.
Response: The study by Bell et al. has been removed - tables have been edited to include only those included in the meta-analysis
Round 2
Reviewer 2 Report
Dear Dr Luis A. Moreno, Section Editor in Chief, and Dr. Miss Ligia Cimpean, Assistant Editor of Nutrients
Thank you for reviewing the manuscript entitled " Effects of Omega-3 Supplementation Alone and Combined with Resistance Exercise on Skeletal Muscle in Older Adults: A Systematic Review and Meta-Analysis" (Manuscript ID: nutrients-1710648 – R1.
GENERAL COMMENTS
I congratulate the authors for the evident improvements in the manuscript. The authors have done a great job of strengthening the present manuscript in a short time. I emphasize that the meta-analysis of each outcome (with and without strength exercise) makes the information much clearer, allowing the reader to broaden their understanding and discussion of the possible benefits of omega-3, as well as the weighted weight of each study in the effect set. This new organization adopted by the authors translates into an objective manuscript.